# OpenReview forum: "A Unified Understanding and Evaluation of Steering Methods"
_TMLR — Withdrawn by Authors_

### Review · Reviewer_KEdd · 2025-06-12

**Summary Of Contributions:**

The paper claims that the current steering methods lack a unified understanding and consistent evaluation across tasks and datasets. In light of this view, the authors propose a unified framework for analyzing and evaluating steering methods, bridging theoretical and practical
perspectives.

**Audience:**

Yes

**Claims And Evidence:**

Yes

**Requested Changes:**

Please see my above weaknesses.

**Strengths And Weaknesses:**

Strengths
1. The overall discussions and experiments are reasonable and mainly comprehensive (see weakness).
2. The paper is easy to follow.

Weaknesses:
1. Some research in steering methods is not covered and discussed. For example, representation tuning [1-3]. Do they belong to mean differences? The paper should make sure the variants of steering methods are comprehensively covered, or it might fall into overclaim.
2. In 3.2, the authors discuss that the mean difference approach minimizes the average error across the entire distribution; it does not guarantee optimal performance for every individual sample. Based on this, no further discussions are included in order to potentially solve this problem (a critical aspect for this paper). The results in 4 continuously verify that mean difference approaches can reach a satisfying performance, which might mislead the readers into thinking that this approach is sufficient for steering control.

[1] Finding alignments between interpretable causal variables and distributed neural representations

[2] ReFT: Representation Finetuning for Language Models

[3] Re-Imagining Multimodal Instruction Tuning: A Representation View

---

### Review · Reviewer_5QcM · 2025-07-07

**Summary Of Contributions:**

This paper presents a unified theoretical and empirical study of steering methods for LLMs. The authors formalize several existing techniques (e.g., Contrastive Activation Addition (CAA), Representation Engineering (RepE), and Inference-Time Intervention (ITI)) under a common contrastive framework. The main theoretical result shows that the mean of differences between positive and negative embeddings is the optimal steering vector under a squared-error objective. Empirical evaluations on multiple-choice and open-ended generation tasks demonstrate that the mean-difference method outperforms PCA- and classifier-based alternatives.

**Audience:**

Yes

**Broader Impact Concerns:**

None.

**Claims And Evidence:**

Yes

**Requested Changes:**

- The experiments should include more recent models to enhance the credibility and relevance of the findings.

- Presentation: (1) The citation style is incorrect in many places and should be fixed (e.g., RepE Zou et al. (2023) → RepE (Zou et al., 2023)); (2) Numerical precision should be consistent, especially in tables; (3) The top-left subfigure overlaps with the adjacent one and should be adjusted for clarity.

**Strengths And Weaknesses:**

## Strengths:

- The paper includes extensive experiments across a range of behavioral tasks, showing consistent performance gains for the mean-difference method.

- The paper also includes ablation studies (e.g., on optimal layer location) and visualizations provide actionable guidelines for using steering methods in practice.

## Weaknesses:

- While the formalization is useful, the theoretical result mostly confirms what prior work implicitly assumes. The paper does not propose new methods, and the empirical findings largely validate what was expected (i.e., that simple methods often work best).

- The citation style is inconsistent and error-prone (e.g., “RepE Zou et al. (2023)”), figures are sometimes cluttered (e.g., overlapping subfigures), and tables exhibit inconsistent numerical precision (e.g., mixing “63.82” and “62” in the same table), reducing readability.

- The evaluation is limited to a single model (LLaMA-7B-Chat, from a relatively outdated model series), with no discussion of generalization to larger scales or alternative architectures.

---

### Review · Reviewer_WwYa · 2025-07-26

**Summary Of Contributions:**

The paper proves that the mean of differences between the embeddings of two class minimizes the pointwise MSE between the positive and negative embeddings. It compares CAA, RepE, ITI steering methods on the datasets proposed in the CAA paper, for multiple choice and GPT-4 rated open-ended generation. Finally, the paper repeats the layerwise analysis of the CAA paper and finds the same result that applying steering to intermediate layers works best.

**Audience:**

No

**Claims And Evidence:**

No

**Requested Changes:**

1. Change the scope of the abstract, introduction and contributions to explicitly state that the contribution of the paper is to perform a subset of the comparison done in [2] cited above across datasets in Rimsky et al. In general, the writing needs to be much more specific about the science done in this paper rather than the use of generic adjectives.

2. Compare newer activation steering methods to truly claim a "unified understanding and evaluation" in mid-2025.

3. Perform a human-annotated study to measure the quality of GPT-4 judgements (and clearly state which GPT-4 model this is).

**Strengths And Weaknesses:**

**Strengths**
1. Comparing CAA, RepE, ITI on the CAA benchmark settings allows a direct comparison which shows that class mean difference (as proposed in CAA) works best.
2.  The finding applying steering on positive examples can sometimes reduce their accuracy is surprising and notable.

**Weaknesses**
1. The paper strongly overstates its contribution. From the abstract or introduction, one might think the paper contributes new theory and a new unified way to evaluate steering vectors. However, [1] already proved the optimality of class mean difference, and [2] already compared these activation steering methods on open-ended generation, classification accuracy etc. showing the same results. The benchmark datasets used aren't new either, they are borrowed from the original CAA paper by Rimsky et al. (2024). I think a super-set of the results in this paper were already shown in [2] 20 months ago, albeit on a single dataset. I encourage the authors to discuss the unique contribution relative to [1, 2] and Rimsky et al. in a significant re-write of the abstract, introduction and overall contributions of the paper.

2. The paper studies a single model, Llama 7B. This questions the relevance of the study to the current state of the art in both language models and steering. Dozens of new methods have shifted the frontier in steering beyond the methods studied here (these are cited, but not compared in the paper), and to claim a unified benchmarking of steering in mid-2025, they should be included, and Gemma3 or Qwen3 models should be used.

3. The paper studies open-ended generation ratings using GPT-4. Which version of GPT-4, there have been multiple updates? How trustable are the model's annotations?

[1] LEACE: Perfect linear concept erasure in closed form, https://arxiv.org/abs/2306.03819

[2] Evaluating hidden directions on the utility dataset: classification, steering and removal https://www.lesswrong.com/posts/JCgs7jGEvritqFLfR/evaluating-hidden-directions-on-the-utility-dataset. Results were included in Figure 12, Appendix B.3 of the Representation Engineering paper (Zou et al. 2025).

---

### Note · Authors · 2025-08-01

**Comment:**

We would like to thank the reviewers for their valuable comments and appreciate the insightful feedback on the rigor of the work. Addressing the limitations you pointed out will require substantial revamping of our experiments. After careful consideration, we have decided to withdraw the paper at this stage.

**Withdrawal Confirmation:**

I have read and agree with the venue's withdrawal policy on behalf of myself and my co-authors.